# Detection and Measurement of Matrix Discontinuities in UHPFRC by Means of Distributed Fiber Optics Sensing

**DOI:** 10.3390/s20143883

**Published:** 2020-07-12

**Authors:** Bartłomiej Sawicki, Antoine Bassil, Eugen Brühwiler, Xavier Chapeleau, Dominique Leduc

**Affiliations:** 1Laboratory of Maintenance and Safety of Structures, Structural Engineering Institute, Swiss Federal Institute of Technology (EPFL), CH-1015 Lausanne, Switzerland; eugen.bruehwiler@epfl.ch; 2COSYS-SII, I4S Team (Inria), Univ Gustave Eiffel, IFSTTAR, F-44344 Bouguenais, France; antoine.bassil@quadric.arteliagroup.com (A.B.); xavier.chapeleau@univ-eiffel.fr (X.C.); 3Quadric, Artelia Group, 14 Porte de Grand Lyon, F-01700 Neyron, France; 4GeM UMR 6183, University of Nantes, F-44322 Nantes, France; dominique.leduc@univ-nantes.fr

**Keywords:** crack detection, crack opening, distributed fiber optic sensors, DIC, UHPFRC, testing, SHM, microcracking

## Abstract

Following the significant improvement in their properties during the last decade, Distributed Fiber Optics sensing (DFOs) techniques are nowadays implemented for industrial use in the context of Structural Health Monitoring (SHM). While these techniques have formed an undeniable asset for the health monitoring of concrete structures, their performance should be validated for novel structural materials including Ultra High Performance Fiber Reinforced Cementitious composites (UHPFRC). In this study, a full scale UHPFRC beam was instrumented with DFOs, Digital Image Correlation (DIC) and extensometers. The performances of these three measurement techniques in terms of strain measurement as well as crack detection and localization are compared. A method for the measurement of opening and closing of localized fictitious cracks in UHPFRC using the Optical Backscattering Reflectometry (OBR) technique is verified. Moreover, the use of correct combination of DFO sensors allows precise detection of microcracks as well as monitoring of fictitious cracks’ opening. The recommendations regarding use of various SHM methods for UHPFRC structures are given.

## 1. Introduction

To tackle the challenges that are in front of civil engineering—such as reduction in carbon footprint with optimized design, proper allocation of scarce resources through the use of engineered structural materials or extension of service duration thanks to deeper understanding of performance of structures—up-to-date methods should be used.

From the construction material point of view, such a developing technology is the Ultra High Performance Fiber Reinforced Cementitious composite (UHPFRC). It allows for design of refined and more slender structures as well as reinforcing and upgrading existing ones. To fully master its performance on both micro- and macroscopic levels, new measurement techniques are needed. 

Such a possibility is given through the development of Distributed Fiber Optics (DFO) sensing techniques. The DFOs, used mostly in the automotive and mechanical industry, have recently found place in the civil engineering field. During the last decade, Fiber Optics (FO) sensors became increasingly popular in Structural Health Monitoring (SHM), and now, they are the second most used sensing technology in this field [1]. These sensors are small-sized, lightweight and resistant to both chemical degradation and electromagnetic interference. In the past few years, it was demonstrated by numerous researchers that DFOs can be used for measurements of strain and crack opening in ordinary reinforced concrete [2,3,4,5] and fiber reinforced concrete [6]. Still, the verification of usefulness of this SHM for UHPFRC is missing, especially considering the unique behavior of this material under tensile action.

For UHPFRC structures, the photogrammetry with digital image correlation (DIC) is now a well-established technique to track and measure cracks [7,8,9]. However, there were not any attempts to monitor crack propagation and measure crack opening using the DFO sensing techniques in UHPFRC structural elements. Schramm and Fischer [10] tested a slab element and a prestressed beam. For the slab without rebars, the externally glued DFOs were able to detect the apparent strain peaks due to microcracking. However, no estimation of crack opening was done. In cases of beam elements under shear action, the pattern of fictitious cracks was observed using DIC. The DFOs were glued on the steel rebars and used to detect the stress peaks in the reinforcement (stirrups) near these fictitious cracks. Similar instrumentation, with DFOs on the rebars and DIC on the surface, was used for investigation of the stress transfer between UHPFRC and reinforcement by others [11]. The DFOs sensors were employed for SHM of a UHPFRC bridge [12] but without attempt to detect discontinuities. 

This paper presents results of an experimental validation of the use of DFOs to detect and measure discontinuities in UHPFRC structures. The full-scale beam is instrumented with DFOs, DIC and extensometers in order to compare their crack monitoring features. The results are discussed from structural and material points of view, with an outlook for the possible use of DFOs for Structural Health Monitoring of UHPFRC structures.

## 2. Materials and Methods

### 2.1. Distributed Fiber Optics Sensing for Discontinuity Monitoring

Recently, DFO sensors were especially used where an urgent need of high number of sensing points appeared. The difference between these systems and traditional long gauge or point sensors is their ability to provide distributed measurements, and thus, simultaneously local and global information [13]. Measurement systems are composed of an interrogator and an optical fiber playing the role of a sensor. These sensors are either embedded into new concrete structures or bonded to the surface of existing ones. Different interrogation units available nowadays are based on the analysis of the Brillouin and Rayleigh backscattered light over the silica optical fiber. The Rayleigh-based systems can perform distributed strain measurements with higher spatial resolutions (<1 cm) than Brillouin-based systems which, on the other hand, can interrogate larger distances (>100 km). 

The DFOs techniques can be used not only to measure strains but also to detect, localize and measure cracks of small openings. DFO sensors allow achieving of an accurate, reliable and quasi real-time crack detection and characterization in concrete structures. They contributed to the detection and localization of cracks in the massive structures, showing supremacy over the short and long gauge sensors (Figure 1a). Actually, any type of discontinuity in the host material, like a crack, can cause a strain localization propagating through the optical fiber layers up to the core of the optical fiber (Figure 1b). 

There are numerous studies on the use of DOFs for detection of crack formation. The sensors, based on measuring losses with Optical Time Domain Reflectometry technique [3,4,5,14,15,16], were very limited in practical applications. Similarly, those based on Brillouin backscattering [6,17,18,19,20] were limited due to their low spatial resolution, affecting their strain sensing accuracy around a crack in the concrete material [21,22]. In fact, the complicated strain distribution and its rapid variation within the spatial resolution decreases the strain measurement accuracy [23]. Later on, Optical Backscattering Reflectometry (OBR), based on the Optical Frequency Domain Reflectometry (OFDR) technique, emerged. This technique, characterized by high spatial resolution, is proven to be capable of detecting and localizing tiny microcracks in reinforced concrete structures [24]. Different methods were also proposed to quantify crack openings from the strain profiles, either based on a combination with finite element models of the structure [25,26] or on the calculation of the optical fiber elongation by summing distributed strain gradients [27,28].

### 2.2. Analytical Models Based on Strain Transfer Theories

A distributed optical fiber sensor is an optical fiber surrounded by various protective and adhesive layers, forming a multilayered strain transfer system. The existence of these intermediate layers leads to differences in the strain of host material and the strain measured by an optical fiber due to the shear lag effect in intermediate layers. The problem of strain transfer through an optical fiber sensor has been studied in the field of short dimensional sensors like Bragg grating or interferometric sensors [29,30,31,32,33,34,35]. Indeed, many research works focused on designing discrete sensors with improved strain transfer efficiency [36] and performing parametric studies of different mechanical and geometrical properties of multilayered sensors [37]. 

Since 2012, different analytical and numerical models were proposed [38] to describe the strain transfer from a discontinuous (cracked) host material. Imai et al. [6] introduced the effect of crack discontinuity in host material as a Gaussian distribution at the interface with protective coating. Later, it was assumed that the strain at the discontinuity location is equal to the crack opening over the spatial resolution of the measurement instrument [39]. Finally, the Crack Opening Displacement (COD) was introduced as an additional term provoked by the local discontinuity in the host material deformation field. Feng et al. [18] deduced a mechanical transfer equation, showing that the strain measured by the optical fiber *ε_f_(z)* consists of a crack-induced strain *ε_crack_(z)* part added to the strain in host material *ε_m_(z)*. Recently, Bassil et al. [40,41] deduced a similar strain transfer equation for a multilayer system with imperfect bonding between layers:(1)$$\varepsilon_{f}\left(z\right)=\varepsilon_{crack}\left(z\right)+\varepsilon_{m}\left(z\right)=\lambda \frac{\mathrm{COD}}{2}e^{-\lambda \left|z\right|}+\varepsilon_{m}\left(z\right)$$
(2)$$\lambda^{2}=\frac{2}{E_{f}r_{f}^{2}\left[\frac{1}{G_{1}}ln\left(\frac{r_{1}}{r_{f}}\right)+\sum_{i=1}^{N}ln\left(\frac{r_{i}}{r_{i-1}}\right)+\sum_{i=1}^{N}\frac{1}{k_{i}r_{i}}\right]}$$
where *λ* is the strain lag parameter that includes mechanical (*G* and *E*, shear and Young’s moduli, respectively) and geometrical properties (*r*) of the different *i* layers and *z* is the position of discontinuity along the optical cable. It also includes coefficients *k* depicting the level of interfacial adhesion between two consecutive layers.

The strain lag parameter *λ* is crucial for the sensing of cracks. As this value increases, the crack-induced strains *ε_crack_(z)* figure higher peaks at the crack location, and thus, the exponential part covers a narrower zone over the optical fiber length, as shown in Figure 2. Thanks to this, the capacity of detecting and localizing discontinuities increases.

The authors also demonstrated the validity of the model for different types of optical cables through an experimental testing campaign on concrete specimens. The estimated CODs proved to be accurate, reaching relative errors of 1–10% for a dynamic range [COD_min_, COD_max_], with a strain repeatability of ±20 µm/m for the interrogator unit. In this range, the layers behave in an elastic manner and sufficient, stable bonding between them exists. COD_max_ varied widely from 80 to 1500 µm for different types of optical cable assemblies. On the other hand, the authors fixed the COD_min_ to 50 µm, below which other parameters prevail, i.e., the nature of cracking of concrete material (in the fracture process zone) or the strain accuracy and repeatability of interrogator. 

In terms of crack detection, Bassil et al. [42] demonstrated that an OBR system with a strain repeatability of ± 2 µm/m and an optical cable with *λ* = 20 m^−1^ can detect concrete discontinuities of less than 1 µm.

### 2.3. Ultra High Performance Fiber Reinforced Cementitious Composite

UHPFRC is a composite fiber reinforced cementitious building material with a high content (>3% vol.) of short (*l_f_* < 20 mm) and slender steel fibers. Its behavior under tensile stress comprises three stages, as shown in Figure 3.

The first stage is an elastic stage. The cementitious matrix is continuous and the behavior of UHPFRC is simply linear. The strain of the material can be directly measured.

After the elasticity limit (*f_e_, ε_e_*) is reached, discontinuities in the matrix start to appear and the material enters the strain-hardening phase. The openings of these fine, distributed microcracks (hairline cracks) are smaller than 50 µm and their spacing can vary from 2 to 30 mm [43,44,45]. They are not detrimental from a durability point of view [46,47] and are impossible to see with the naked eye. These microcracks can be, however, measured using appropriate instrumentation. From the macroscopic point of view, the material can be considered as continuous, with strain-hardening quasi-linear behavior and reduced stiffness [48]. However, after unloading, the residual strain remains in the material.

When the maximum tensile resistance *f_u_* is reached, the material enters the softening phase. One or more neighboring microcracks start rapidly growing, eventually reaching openings above 50 µm. This localized discontinuity is bridged by multiple fibers carrying the tensile stress. It is called a fictitious crack, contrary to the real crack which cannot transfer the stress [49]. Since the stress transfer capability in this critical zone is reduced, the overall stress in the area decreases. The localized fictitious crack is growing, while the strain and stress around it decrease. The location of the fictitious crack depends on the distribution of fibers [44,45,50,51]. Since this fictitious crack leads eventually to the failure of structural elements, it is called the critical crack as well. With the gradual opening of the fictitious crack, the measured deformation increases, leading to fast growth of the apparent strain. Importantly, it is not the real strain of the material anymore due to the fictitious crack localized between the reference measurement points.

The fictitious crack grows until opening of half of the steel fiber length, in the present case *l_f_*/2 = 6.5 mm [44] and the resistance of the material decreases. After the fibers are pulled out, no more stress transfer is possible and the real crack is formed.

In the case of a structural R-UHPFRC (Reinforced UHPFRC) element under bending action, the tensile behavior of UHPFRC has important influence on the overall response. Each of the three stages are present under the maximum bending moment in the critical section, where the fictitious crack forms. Under the assumption that the cross-sections remain plain, the distribution of stress along the height of the beam is nonlinear due to nonlinearity of the constitutive law of UHPFRC, as presented schematically in Figure 4. The proportion between parts in elastic, strain-hardening and softening regimes depends on geometry of the element [44].

## 3. Test Set-Up and Specimen

The tested beam has a T-shaped cross-section and dimensions according to Figure 5. This kind of design refers to the use of UHPFRC for waffle deck or unidirectional ribbed slab designs. An example of such a structure is the railway bridge in Switzerland described in reference [52].

Commercially available UHPFRC premix Holcim710^®^ was used, with 3.8% vol. of 13 mm long straight steel fibers with an aspect ratio of 65. Its mechanical properties as obtained by material testing were compressive strength *f_c_* = 149 MPa; elastic limit stress *f_e_* = 6.3 MPa; tensile strength *f_u_* = 12.0 MPa; strain at *f_u_* − *ε_u_* = 3.5‰; modulus of elasticity *E* = 41.9 GPa. Steel reinforcement bars B500B (with *f_sk_* = 500 MPa) were used for both stirrups and longitudinal rebars.

To observe the behavior of UHPFRC at the three stages of its performance, the beam was designed with a reinforcement bar of 34 mm diameter at a height of 187 mm from the bottom, thus, the distance between the bottom face of the beam and bottom of the reinforcement was 170 mm at midspan (Figure 5), allowing for observation of unreinforced UHPFRC. To impose bending failure rather than shear failure under four-point bending, Ω shaped stirrups, Ø 6 mm, were placed outside the constant bending moment zone. Additionally, L-shaped Ø 34 mm reinforcement bars were used on the bottom of the beam, outside of the constant bending zone, to increase the lever arm of longitudinal reinforcement in shear.

The beam of 2 m span was tested in displacement-controlled four point bending. The displacement of the servo-hydraulic actuator was transmitted with the use of hinges and a steel beam. The application points were symmetrically positioned at ±0.25 m from the midspan of the R-UHPFRC beam. The course of the actuator was done with velocity of 0.01 mm/s during the first loading, and 0.02 mm/s in unloading and re-loading phases. Several unloadings were performed to obtain the residual deflection of beams at each loading stage.

The beam was instrumented with extensometers; photogrammetry DIC by means of two 20 MP cameras; DFO sensors (Figure 5). The fiber optics sensors for distributed strain sensing were installed in three lines at each face of the beam—40, 90 and 190 mm from the bottom of the beam. As shown in Figure 6, the DFO sensors were glued in a 2 × 2 mm groove on the UHPFRC surface using a bicomponent epoxy adhesive (Araldite 2014-2). On the front side of the beam, the SMF-28 Thorlabs^®^ fiber was glued, with an external diameter of Ø 900 µm and elastomer tubing. On the back side of the beam, the Luna^®^ High-Definition Polyimide fiber was used, with an external diameter of Ø 155 µm. The DIC measurement zone spanned 35 cm from the midspan symmetrically, and over the whole height of the beam on the front side. The extensometers of 100 mm measurement base were glued on the back side of the beam, at the level of each DFO measurement line. Additionally, three LVDTs with the common measurement base were vertically installed on the back side of the beam, at midspan and over the supports. The mean vertical displacement over the supports is subtracted from the vertical displacement at midspan to obtain the deflection of the beam. The resistance force was measured by the load cell of the actuator.

## 4. Test Results

### 4.1. Global Response of the Beam

The force–midspan deflection curve is presented in Figure 7. Several loading–unloading cycles were executed at different stages of the test. The goal was to visualize the influence of residual strain of the UHPFRC in the strain-hardening domain after unloading on the global response of the beam. Thanks to this, a gradual degradation of material can easily be observed. The load steps (LS) were chosen arbitrarily to discuss the state of material in detail.

The first linear part of the curve is very short. This is due to the material at the bottom of the beam entering the strain-hardening regime relatively soon. As the zone where UHPFRC is in the strain-hardening regime is growing, gradual reduction in material stiffness, and thus, beam rigidity occurs, effecting nonlinearity of the force–deflection curve. The residual deflection in the unloading cycle comes from the fact that this part of cross-section contains discontinuities (microcracks < 50 µm) or, in the further stages, the fictitious crack (> 50 µm) is present. Both types of discontinuities transmit the tensile stresses thanks to fibers, but do not close completely while unloaded. Finally, when the beam resistance is maximum with the force of 313 kN, gradual degradation with a rise in deflection continues as the localized fictitious cracks propagate and the longitudinal rebar is yielding.

### 4.2. Detailed Examination: DIC

The unloaded state of the beam is presented in Figure 8. The measurement noise for DIC is mostly at the level of ±100 με, with local peaks of −300 με. The strain noise is not only due to the quality of the cameras and nonuniformity of light, but mostly from the variation of size and distribution of speckles.

Figure 9 presents the horizontal (*ε_xx_*) strain distribution measured with DIC at different load steps (LS), marked in Figure 7. The color scale is the same as presented in Figure 8. LS 1 is not shown since mostly the noise is registered. At load step 2, the uniform elastic strains are registered. At LS 3, strain peaks are observed due to the distributed microcracking of UHPFRC in strain-hardening. They are more pronounced in two weak spots at around −0.25 m and +0.25 m (Figure 9b). While the microcracks keep growing and propagating, two of them grow faster than others, leading to localization of fictitious cracks (Figure 9c). At LS 5, fictitious Crack 1 develops a second, left branch. This could be due to the first branch reaching a stronger area with higher concentration of steel fibers (Figure 9d). Simultaneously, fictitious Crack No. 2 keeps propagating on the right side. At LS 6, the fictitious cracks are clearly visible to the naked eye (Figure 9e). As the fibers bridge these macrocracks, the overall response of the beam remains in the hardening domain. When the beam reaches the peak resistance with a force load of 313 kN, the fictitious cracks reach the level of the reinforcement bar (see Figure 9f). In their bottom part, they transmit hardly any stress due to the large opening and advanced fiber pullout. This is why the bottom part of the beam between the fictitious cracks is almost unloaded. The highest strains are present at the level of the reinforcement bar. After this stage, due to transformation of the fictitious cracks into real cracks with no stress transfer and reinforcement yielding, the resistance of the beam started to decrease and the test was stopped.

### 4.3. Detailed Examination: Strain Measurements

The strain is directly obtained from DFOs and extensometers. For DIC, the virtual measurement lines (Figure 8), positioned at the same height as the DFOs and extensometers (Figure 5), were prepared in the post-processing software VIC 3D^®^. 

The measurements taken along each measurement Line 1, 2 and 3 are presented below from the top to the bottom, respectively, in coherence with their position over the height of the beam (Figure 5).

At load step 1 (LS 1), all systems, except DIC, show good agreement (Figure 10). The strain spatial distribution can be considered as uniform in the constant bending moment zone, between ±0.25 m from the midspan. The material remains elastic and the structural response is linear. In the bottom line (Line 3), local variations of modulus of elasticity can be visible with extensometers and Polyimide, but not with Thorlabs fiber. Possibly, initial microcracks appear. Two strain peaks are visible at Line 2 position −0.25 m. Apparently, there is a local defect of the UHPFRC material there, possibly due to the fabrication of the beam. It can be considered as a disturbance on the surface, since other measurement techniques do not record it.

At LS 2 (Figure 11), as UHPFRC enters into the hardening stage at Lines 2 and 3, clear peaks from microcracks are visible over the Polyimide lines, while Thorlabs do not present any local strain variations. The microcracked zones can be identified with extensometers as well, which show higher strains than in the neighboring zones. It is important to mention that at this stage, propagating microcracks are identifiable by the naked eye when the surface is wetted with alcohol. The difference between the measurements of systems deployed on the back (Polyimide, extensometers) and the front (Thorlabs, DIC) sides of the beam may come from non-horizontal loading of the beam, despite the hinge between the actuator and redistribution beam, or to locally weaker material close to the surface. In perfect conditions, the total elongation obtained with extensometers and DFOs, thus, ‘smeared’ strain, are equal [39]. These incoherencies are not observed for Line 1, which remains in the elastic stage.

Under the force of 90 kN (LS 3, Figure 12), more microcracks are visible over Line 3. Most of these cracks are localized around positions −0.2 m and +0.25 m. Line 2 presents more uniform microcracking behavior. The origin of this nonuniformity in the bottom of the beam is discontinuity of the L-shaped rebars (see Figure 5) causing disturbance of fiber orientation and concentration of stresses. Zones where the fictitious crack will further develop are now clearly visible with DIC and Thorlabs fiber (Crack 1 and 2 line 3, Crack 1 line 2), as well as extensometers (both fictitious cracks, Lines 2 and 3). For zones where the fictitious cracks are developed, the apparent strain measured with extensometers cannot be considered as material strain anymore (see Figure 3). Clear microcracks start to appear at Line 1.

At LS4 (Figure 13), dropout points start to appear at crack locations in Lines 2 and 3 in the DFOs results. These points are dropped out by the spectral shift calculation algorithm due to low correlation with the reference spectrum. This phenomenon of miscalculated points increases due to rapid variation of strain over the spatial resolution length. For fictitious Crack 1, Line 3, DIC shows three fictitious crack fronts forming at positions: −0.25, −0.2 and −0.17 m. They all lay within the same extensometer measurement base, and thus, cannot be distinguished with this technique. Additionally, the Thorlabs fiber is arguably not sensitive enough to clearly separate these fronts due to low shear lag parameter *λ* (Equation (1)). For fictitious Crack 2, Line 3, the apparent strain reaches *ε_u_*, exponential shape is being formed in Thorlabs, and UHPFRC enters softening stage. Two other fictitious crack fronts can be noticed with DIC but hardly with Thorlabs. Extensometer of location [0.05; 0.15 m] does not show fictitious crack formation, while it is visible in the same position with DIC and Thorlabs fiber. This comes from the nonorthogonality of the crack regarding the beam axis and is confirmed by Polyimide fiber recording only microcracks in the discussed location. On Line 2, localization of fictitious Crack 2 starts being detectable by Thorlabs fiber and DIC.

At LS 5 (Figure 14), both fictitious cracks are clearly formed in Line 3, and UHPFRC is in the softening stage. The DFOs do not work properly in their vicinity anymore. The transversal skewness of fictitious Crack No. 1 can be seen, since the peak of DIC is shifted with respect to the extensometers. Interestingly, it is positioned some 7 cm towards the left regarding the previously observed strain concentration. For both Lines 2 and 3, the clear exponential shapes can be noticed in Thorlabs fiber measurements, but with multiple dropouts. While comparing the measurement Line 1 at the current load step with Line 2 and Line 3 at LS 3, it can be concluded that microcracking is more uniformly spaced for the lines positioned higher on the beam. The reasons might be the nonuniformity of fiber dispersion and discontinuity of strains, both due to the rebar alignment. At this load step, the fictitious cracks are clearly visible to the naked eye, and UHPFRC is in the softening stage (see Figure 9d). The stress transferred by bridging fibers is lower than *f_u_* (see Figure 4), and stress in the neighboring material decreases. Thus, the strain measured at midspan is similar for all the measurement lines.

Due to multiple dropouts, DFOs are not useful anymore. The DIC measurements were presented before and, as mentioned above, the results obtained with extensometers crossed by fictitious cracks are not useful. Thus, the detailed analysis of strains ends here.

### 4.4. Monitoring of Fictitious Crack Opening

After examining the DFOs, discontinuity detection performance, it is interesting from both a structural and material point of view to follow the material discontinuities that evolve to discrete fictitious cracks in order to assess their implication on the safety of the UHPFRC structure. Thus, in this section, the strain transfer model is applied to Thorlabs fiber measurements. The Polyimide fiber was not examined because of its limited dynamic range that does not exceeded 80µm in ordinary concrete [41], preventing fictitious COD monitoring. 

The notation of COD is continued here in view of previously discussed state of the art for crack measurement in concrete. As explained before, UHPFRC has more complex response under tensile action. Conveniently, the term COD refers to opening of the matrix discontinuity, be it a microcrack in strain-hardening stage, a fictitious crack bridged by fibers or a real crack with no stress transfer.

The mechanical strain transfer equation for the multiple cracks case is fitted to the strain profiles using the robust least square method:(3)$$\varepsilon_{f}\left(z\right)=\overset{20}{\underset{i=1}{\sum }}\frac{{\mathrm{COD}}_{i}}{2}\lambda e^{-\lambda \left|z-z_{i}\right|}+\varepsilon_{m}\left(z\right)$$
where COD_i_ is the opening displacement of each discontinuity *i*, and *λ* is the strain lag parameter. Each COD_i_ and *λ* are selected as variable parameters. Similar to [42], a trapezoidal approximation of material strain *ε_m* (*z*) is adapted based on the measurements outside the constant bending moment zone; *z_i_* corresponds to the position of the 20 most important strain peaks in the strain profiles.

Figure 15 and Figure 16 present fitted strain profiles to those measured over Line 2 and Line 3 respectively, together with the corresponding residuals for different load levels. A discontinuity propagates in the UHPFRC material through searching the lowest energy path depending on the local fiber content and orientation [51]. Despite the host material’s complex microcracking nature, the proposed mechanical model fits clearly the distributed strain profiles measured by the DFOs system at different levels. Low residual levels are randomly scattered around zero all over the length of FO Line 2 and 3.

On the left beam part, two microcracks are developing to form fictitious cracks. Unlike in concrete, there is no immediate unloading around these discontinuities. Thus, when the fictitious crack localizes and the stress transferred by bridging fibers reaches the value *f_u_*, another fictitious crack can appear nearby. This phenomenon is observed with fictitious Crack 1, where the propagation of one branch stops (Crack 1-Right) and a second one develops (Crack 1-Left). On the other hand, fictitious Crack 2 goes through a more localized propagation. When the force reaches 170 kN for Line 3 and 200 kN for Line 2, an increase in strain residuals is observed around Crack 2. As discussed in reference [41], this could be attributed to the optical cable/host material mechanical system entering a post-elastic phase.

Figure 17 shows the estimated strain lag parameter *λ* as well as the discontinuity openings COD_i_ of fictitious Cracks 1 and 2, under loading and unloading cycles. For both Line 2 and 3, the estimated strain lag parameter *λ* varies around 35 m^−1^ in a ± 10% interval (Figure 17a,c). Higher *λ* values can be observed at early stages of the tests. Similar to previous findings on concrete structures [42,53], this variation can be associated with the first stages of UHPFRC cracking behavior, where discontinuities in the cementitious matrix are starting to develop in the so-called fracture process zone, and end up leading to the creation of a microcrack. When most of the matrix discontinuities exceed an estimated opening COD_min_ of 50 µm, the strain lag estimations become stable and consistent around λ ≈ 35 m^−1^. This confirms the assumption of one global strain lag parameter characterizing the Thorlabs fiber/epoxy glue/UHPFRC mechanical response in the presence of a fictitious crack. Lower *λ* (compared to concrete’s surface-mounted fibers (50 m^−1^)) can be attributed to a lower stiffness level at the Epoxy/UHPFRC interface, possibly due to much smaller porosity.

The estimated CODs for discontinuities Cracks 1 and 2 are shown in Figure 17 c and d. At the level of Line 3, the discontinuities Crack 1-right and Crack 2 are formed as microcracks (<50 µm) and propagate steadily until a force of around 80kN, where they grow rapidly to form fictitious cracks. At F = 120 kN, another microcrack grows rapidly to form the fictitious Crack 1-left. This growth leads to a decrease in the growth rate of fictitious Crack 1-right.

Similar development of COD for the three discontinuities can be observed for Line 2, with a delay regarding Line 3 due to its closer position to the neutral beam axis. Akin two-phased growth of COD is observed: stable during microcracking and fast once the fictitious crack is formed in the softening phase. 

The growth of COD of fictitious Crack 2 is faster than Crack 1, where the damage development is shared by the two branches. Once it reaches a COD_max_ of 400 µm, unstable growth in estimated COD is observed in both measurement lines. This threshold marks the validity limit of the strain transfer model, where all the layers behave in an elastic manner with no progressive debonding occurring at successive layer interfaces. This phenomenon, equally observed in concrete [41], is pronounced by a change in the exponential form of the strain profiles initiated near the strain peak, and thus, leading to an increase in strain residuals (Figure 15 and Figure 16). Consequently, this leads to a change in the tendency of *λ* and COD variations due to increased estimation errors.

Importantly, the COD and *λ* estimations show proper agreement for the loading–unloading cycles. In other words, the UHPFRC as well as the optical fibers attached to it deform in the same manner, even under multiple crack opening and closing over an important area of the beam. It also shows the great potential of the DFOS techniques to monitor residual and periodical openings of discontinuities, which is an important feature for long-term structural health monitoring and studying of the fatigue of the structural elements. 

In this experiment, the large noise level of DIC measurement prevented accurate microcrack and early fictitious crack opening measurements. The COD values obtained using DIC were outside of applicability of the DFOs measurement method. Thus, the results cannot be validated using both methods.

## 5. Discussion

The detailed analysis of results revealing the differences in performance of discussed measurement techniques is presented and summed up in Table 1.

The extensometers allow for measurement of strains in the elastic and strain-hardening phases, which is important from a practical ‘smeared’ approach point of view. They allow for early detection of microcrack propagation with faster rise of strains in the given area in the strain-hardening phase of the UHPFRC response. However, it is impossible to distinguish between accumulation of distributed microcracks and the onset of the fictitious crack formation. Thus, the determination whether the material is in the hardening or softening phase cannot be directly achieved. Additionally, the strain resolution and the localization of discontinuities is limited to the measurement base length of the extensometer. Furthermore, they do not allow for measurements of the fictitious crack opening. Still, they remain the measurement technique that is the easiest in installation and provide results that can be analyzed straightforwardly.

Due to the large measurement noise, the DIC technique did not allow in this experiment for observation of strain variations during the elastic stage of the structural response. However, it allows for tracking the localized fictitious cracks, particularly their length and their opening, at the macro-level. The large measurement noise is due to the relatively large measurement field (0.7 × 0.4 m) and nonuniformity of speckles. It was proven that this technique allows for tracking of microcracks for smaller observation fields [45]. This method remains highly complex and sensitive in practical application for Structural Health Monitoring.

The results obtained using the DFOs technique depend on the sensitivity of the used optical cable or fiber, with regard to discontinuities in the host material. The fiber with Polyimide coating features high sensitivity, allowing early and accurate detection and localization of microcracks. Through computation of the total elongation of segment of fiber, the strain of UHPFRC in the strain-hardening domain can be obtained [28]. Therefore, both the practical ‘smeared’ as well as ‘discrete’ approach to distributed microcracking can be used. This is relevant for Structural Health Monitoring, as structural UHPFRC remains in the elastic or strain-hardening state during normal service life. 

On the other hand, the Thorlabs fiber with Acrylate and Hytrel double coatings features lower crack sensitivity than Polyimide fiber. This allows for strain measurement during elastic and strain-hardening stages. The detection, localization and measurement of microcracks is limited due to its sensitivity. It is however capable of detecting and localizing fictitious cracks, as well measuring their opening since their formation and up to 400 µm. More importantly, in this range, the optical fiber sensors maintain their elastic behavior allowing accurate estimation of cracks widths during closing–opening cycles. From a practical point of view, formation of fictitious cracks can indicate problems in the UHPFRC structure, for example, due to overloading. Thus, this kind of DFOs can play an important role in SHM and verification of structural safety. 

In order to take full advantage of the DFOs technique, both types of optical fibers with their different crack sensitivity could be used to monitor the behavior of UHPFRC in the elastic, strain-hardening and softening domains. 

In recent years, rapid development in the field of DFOs interrogation units enabled accurate, continuous, dynamic and simultaneous strain sensing along different types of optical fibers. With a better understanding of the sensor properties (like crack and temperature sensitivity) and durability (long-term fatigue and aging), DFOs technique can perform global and local strain measurements to provide information on the overall UHPFRC behavior in a holistic manner. Thus, DFOs can form an undeniable asset for long-term continuous health monitoring of this type of new structures.

## 6. Conclusions

In this work, the DFOs technique based on the Rayleigh backscattering phenomenon is used to follow the behavior of the R-UHPFRC beam tested under four-point bending. The capacity to measure strains and monitor matrix discontinuities with two types of optical fiber sensors was evaluated. The comparison with DIC and extensometers revealed application ranges of each method. 

The usefulness of extensometers is limited to the elastic and strain-hardening phases. They can measure strains in the UHPFRC and detect microcrack accumulation. It is impossible to distinguish between advanced microcracking and nucleation of fictitious cracks. 

The DIC is highly dependent on size of the measurement field. In this research, it allowed for detection and tracking of fictitious cracks. The complexity regarding the measurement area preparation and data processing makes this technique too complex to be used in situ for now.

The DFOs technique is able to precisely monitor the elastic, strain-hardening and softening stages of UHPFRC. While using a high spatial resolution OBR measurement technique, its performance depends on the type of fiber used for sensing. While strain measurement in the elastic phase or detection and localization of microcracks is of interest, Polyimide coated optical fiber should be used. If the strain measurement in both elastic and strain-hardening phases or fictitious crack detection and localization is to be observed, the Thorlabs fiber with thicker coating prevails.

It was confirmed that the COD of fictitious cracks can be successfully estimated using the proposed analytical model with proper choice of sensing optical fiber. Importantly, the coherent estimation of opening–closing fictitious crack width shows the potential of this method for SHM under repeated loading and real-time SHM of UHPFRC structures. However, testing of the optical fiber sensors under high numbers of crack closing/opening cycles should be considered.

## Figures and Tables

**Figure 1 sensors-20-03883-f001:**
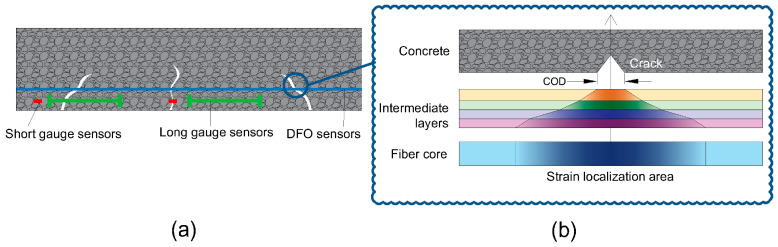
Crack detection in concrete using Distributed Fiber Optics sensing techniques; (**a**) comparison to traditional sensors, (**b**) strain transferring between layers.

**Figure 2 sensors-20-03883-f002:**
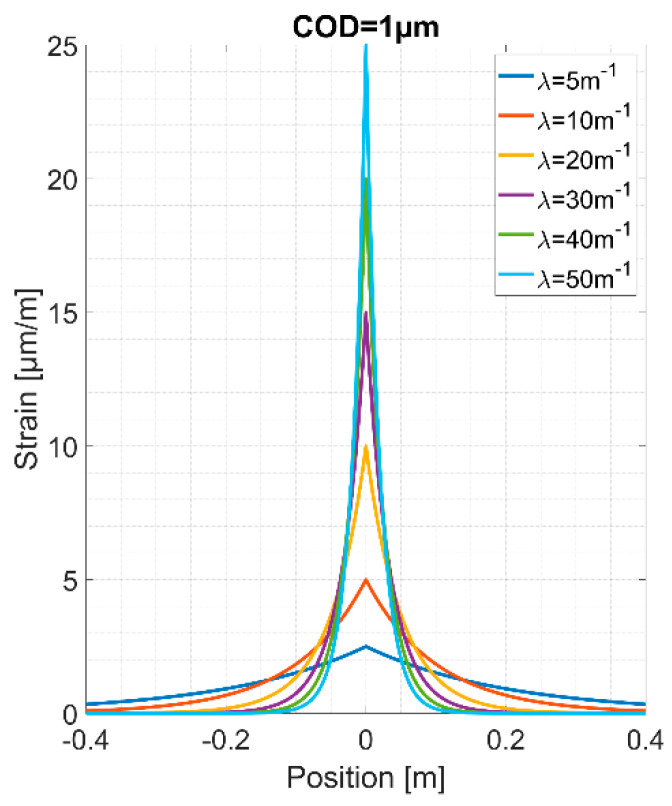
The spatial distribution form of the crack-induced strains *ε_crack_(z)* for different strain lag parameter *λ* values.

**Figure 3 sensors-20-03883-f003:**
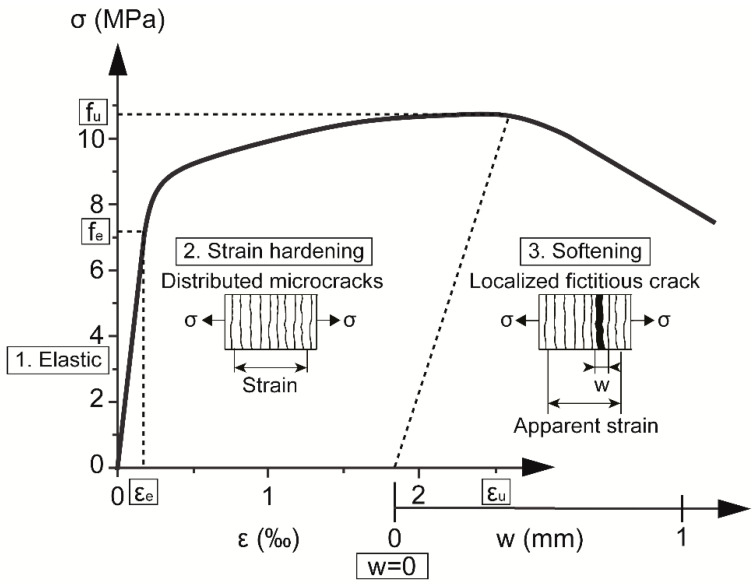
UHPFRC behavior under tension.

**Figure 4 sensors-20-03883-f004:**
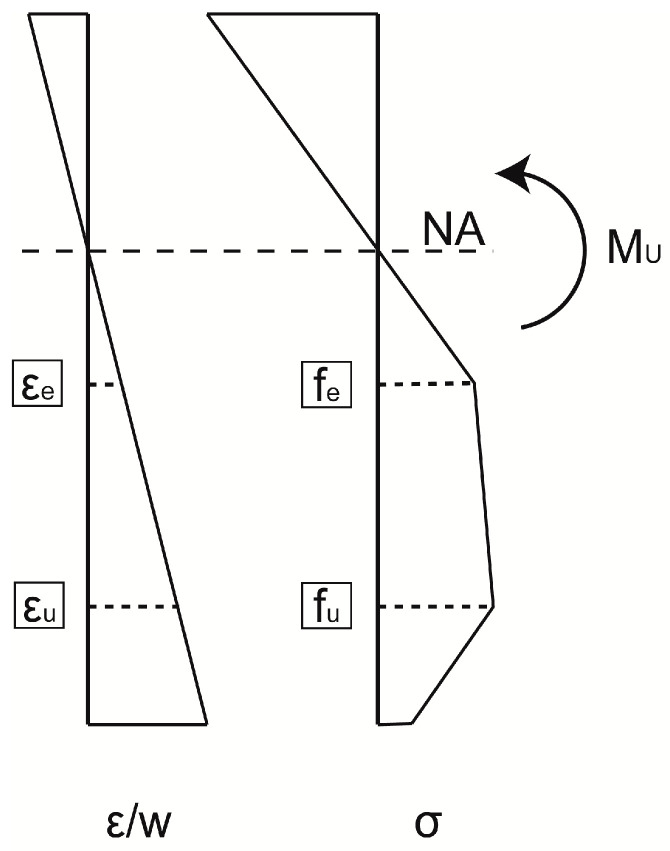
Schematic distribution of stresses and strains in UHPFRC at the critical section of R-UHPFRC beam under the ultimate bending moment.

**Figure 5 sensors-20-03883-f005:**
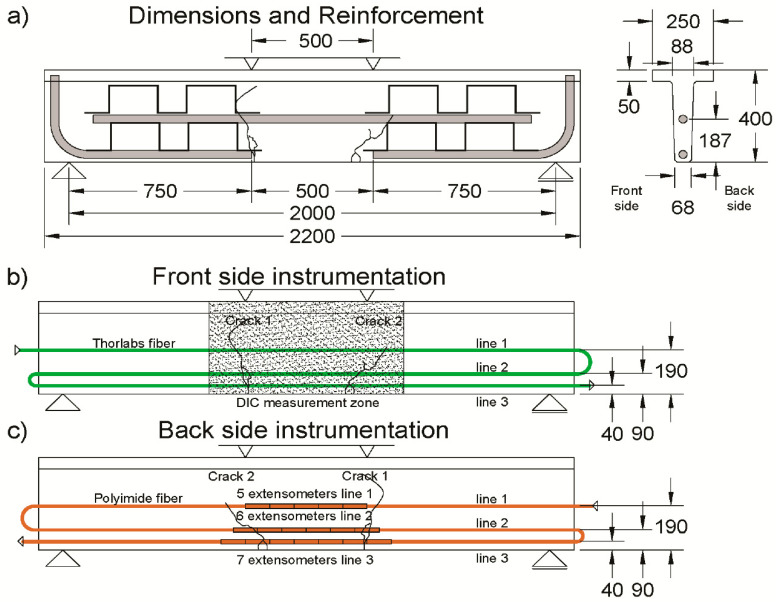
Scheme of test setup and instrumentation: (**a**) beam dimensions and reinforcement, (**b**) instrumentation on the front side and (**c**) instrumentation on the back side; dimensions in mm; the critical fictitious cracks 1 and 2 are marked.

**Figure 6 sensors-20-03883-f006:**
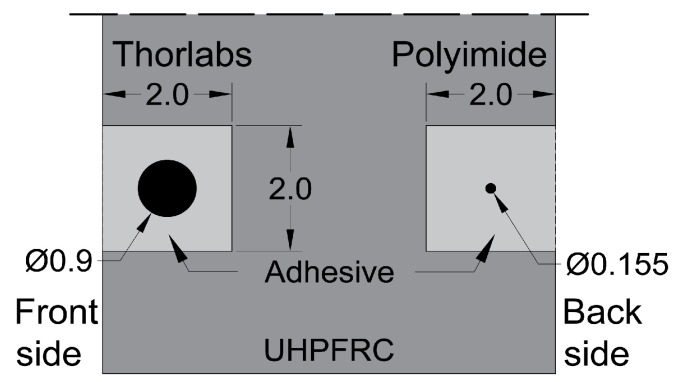
Scheme of installation method of DFO sensors in UHPFRC; dimensions in mm.

**Figure 7 sensors-20-03883-f007:**
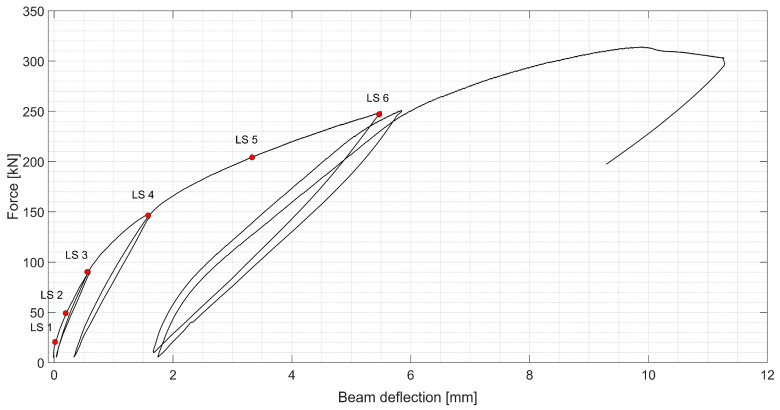
Load–deflection curve of the R-UHPFRC beam during quasi-static test, total jack force presented with consecutive load steps (LS) marked.

**Figure 8 sensors-20-03883-f008:**
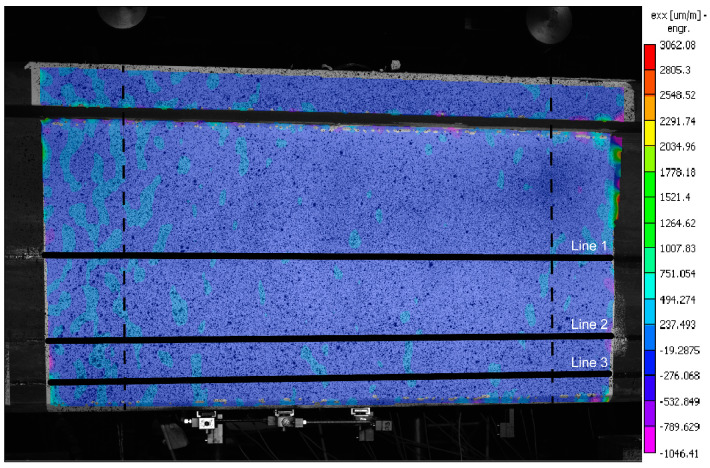
DIC measurement at unloaded state with virtual measurement lines noted; the constant bending moment zone is marked with dashed lines.

**Figure 9 sensors-20-03883-f009:**
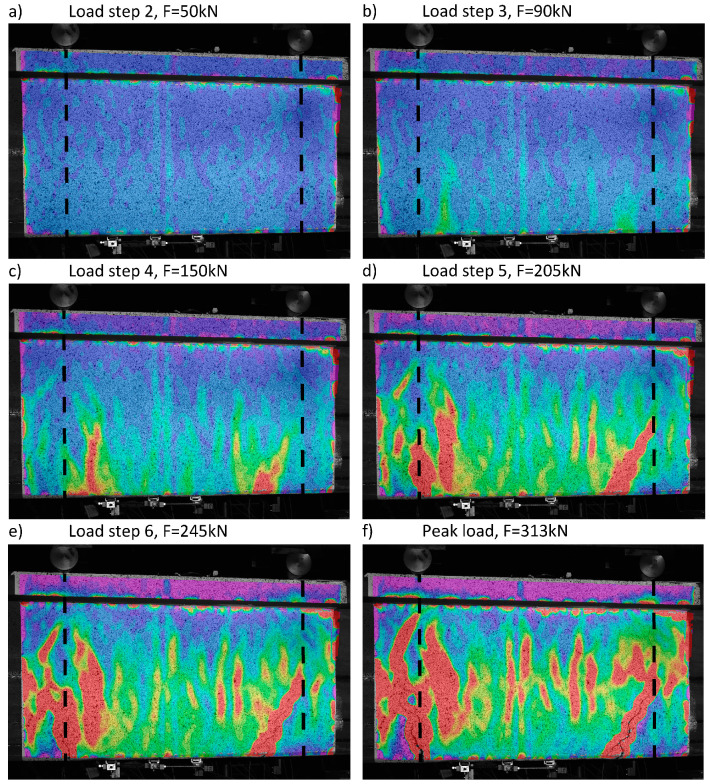
Strain distribution obtained with DIC for different load steps: (**a**) LS2, (**b**) LS3, (**c**) LS4, (**d**) LS5, (**e**) LS6, (**f**) Peak load; ε_xx_ with scale as in Figure 8; the constant bending moment zone is marked with dashed lines.

**Figure 10 sensors-20-03883-f010:**
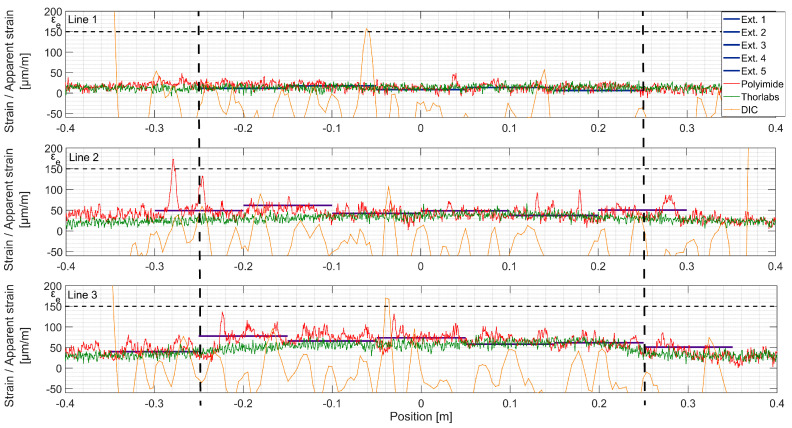
Strains measured with different techniques, load step 1, F = 20 kN; the constant bending moment zone is marked with dashed lines.

**Figure 11 sensors-20-03883-f011:**
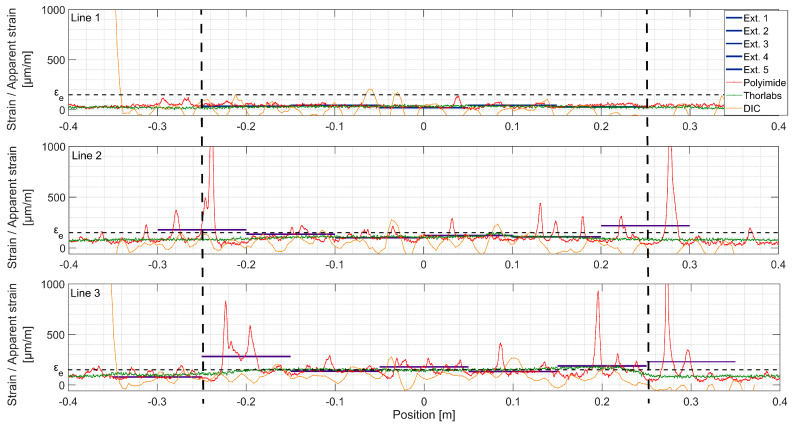
Strains measured with different techniques, load step 2, F = 50 kN; the constant bending moment zone is marked with dashed lines.

**Figure 12 sensors-20-03883-f012:**
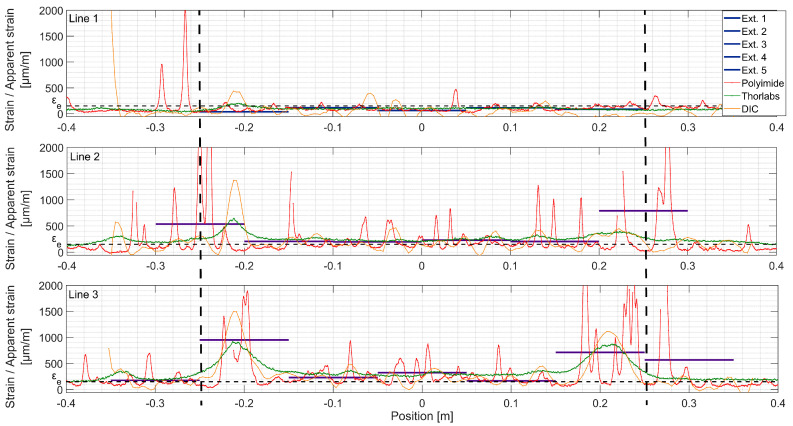
Strains measured with different techniques, load step 3, F = 90 kN; the constant bending moment zone is marked with dashed lines.

**Figure 13 sensors-20-03883-f013:**
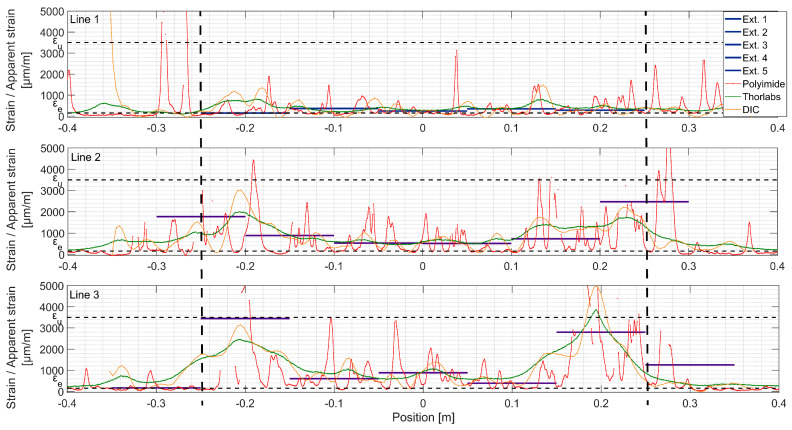
Strains measured with different techniques, load step 4, F = 150 kN; the constant bending moment zone is marked with dashed lines.

**Figure 14 sensors-20-03883-f014:**
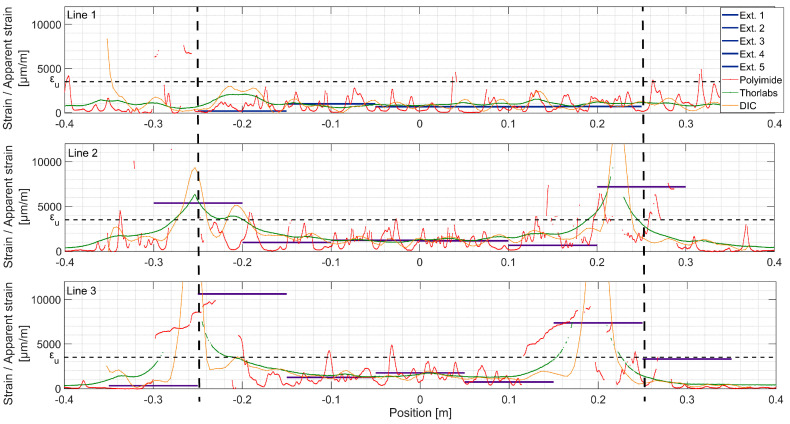
Strains measured with different techniques, load step 5, F = 205 kN; the constant bending moment zone is marked with dashed lines.

**Figure 15 sensors-20-03883-f015:**
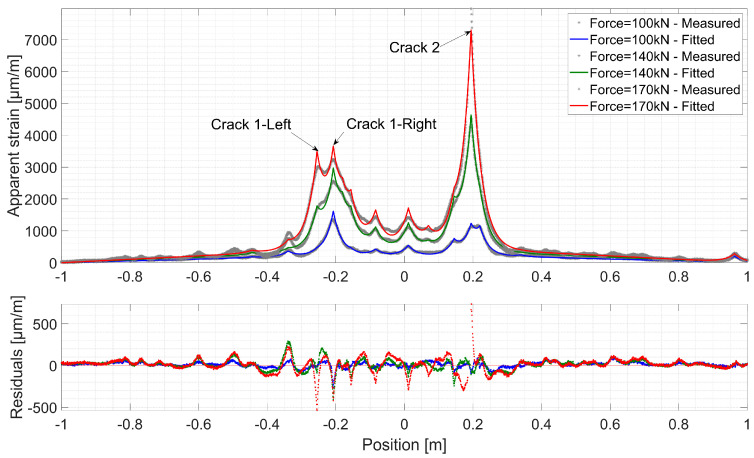
Comparison between the measured and fitted strain profiles and the corresponding residuals over Line 3 of Thorlabs cable.

**Figure 16 sensors-20-03883-f016:**
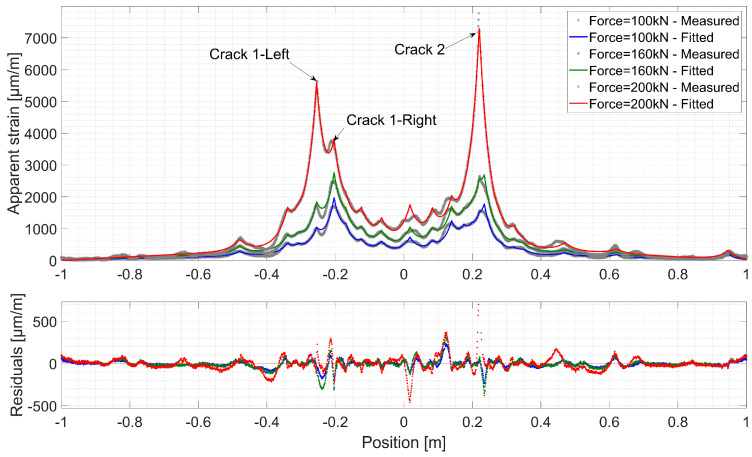
Comparison between the measured and fitted strain profiles and the corresponding residuals over Line 2 of Thorlabs cable.

**Figure 17 sensors-20-03883-f017:**
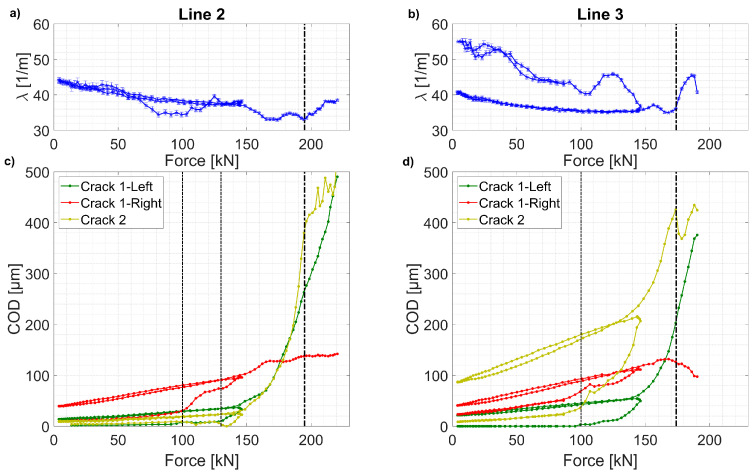
Variation of the estimated COD ((**a**,**b**)) and *λ* ((**c**,**d**)) during the test; fictitious crack initiation marked with thin dashed lines; loss of precision marked with thick dashed lines.

**Table 1 sensors-20-03883-t001:** Comparison of used measurement methods regarding their application range.

Measurement Method		Extensometers	DIC	DFOs	
				Polyimide	Thorlabs
**Strain measurement**	Elastic	**+**	**+**/−	**+**	**+**
	Strain-hardening	**+**	**+**/−	−	**+**
**Distributed microcracks**	Detection	**+**	**+**/−	**+**	−
	Localization	−	**+**/−	**+**	−
	Measurement	−	**+**/−	**+**	−
**Localized cracks / fictitious cracks**	Detection	**+**/−	**+**	**+**	**+**
	Localization	−	**+**	−	**+**
	Measurement	−	**+**	−	**+**
**Comments**		Limited area covered; simplest in application and analysis	Highly dependent on noise and area of interest	Measurement of microcracks theoretically possible	Crack measurement range limited to 400 µm for UHPFRC

**+** - yes / − - no

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
