# Peer review of "Detection and Measurement of Matrix Discontinuities in UHPFRC by Means of Distributed Fiber Optics Sensing"

_sensors, 2020, doi:10.3390/s20143883_

Round 1
Reviewer 1 Report
The authors explain very well their work, the experiments and the results.
However, I not really feel that the paper can be accepted for publication. The reason for this is that is not clearly stated what is the innovation of this work, what they achieved and what improved compare to previous works.
Also, some minor comments are listed below,
Section 2.2: remove the comma from the title
Section 2.3: Since is a title I suggest to use the full name of the acronym.
Line 221: I prefer to start with "In this section..." instead of using the word chapter. line 232 the same...
I will suggest adding a chromatic scale for fig9.
The authors need to explain why they used two fibers with different characteristics.
Author Response
The authors would like to express their gratitude to reviewers for their time spent on this task. Their comments and suggestions were very useful and undoubtedly will help in improvement of the manuscript.
The detailed responses to the comments are listed below. The word file was edited with change tracking to ease-up further stages of verification. The modifications are listed below.
However, I not really feel that the paper can be accepted for publication. The reason for this is that is not clearly stated what is the innovation of this work, what they achieved and what improved compare to previous works.
It is the first time the Distributed Fibrer Optics sensing (DFOs) with Optical Backscattering Reflecterometry (OBR) is used for detection and estimation of discontinuities in Ultra High Performance Fibre Reinforced Cementitious composite (UHPFRC) under external loading. The appropriate comments are added in the introduction and conclusion sections to underline this novelty.
Also, some minor comments are listed below,
Section 2.2: remove the comma from the title
Corrected
Section 2.3: Since is a title I suggest to use the full name of the acronym.
Modified
Line 221: I prefer to start with "In this section..." instead of using the word chapter. line 232 the same...
The paragraph deleted to avoid misunderstanding
I will suggest adding a chromatic scale for fig9.
For sake of simplicity the same scale is used for figs. 8 and 9. The caption of fig. 9 was edited for clarification.
The authors need to explain why they used two fibers with different characteristics.
Explanation added in section 2.2, clarifications in section 5 and 6 (see list of changes)
List of modifications (line numeration of edited article used):
- Whole document: re-ordering of references according to modifications is text
- line 13: typo corrected
- lines24-30: stylistic corrections
- lines 34-68: stylistic corrections
- lines 49-51: sentence added to show novelty of research
- lines 61-63: added reference [12] with use of DFOs for SHM in UHPFRC structure
- line 70: deletion of abbreviation form section title
- lines 71-74: stylistic corrections
- line 94: stylistic corrections
- lines 103-110: addition of references [29-37] on strain transfer in optical fiber sensor
- lines 112,116: stylistic corrections
- line 121: definition of G and E added
- lines 125-127,132,133,135,137: stylistic corrections
- line 143: replacement of abbreviation with full name in the section title
- lines 210, 211,219, 220: stylistic corrections
- line 227: deletion of linking paragraph
- line 232: deletion of “in the following chapters” to avoid misunderstandings
- line 271 (caption fig. 9): replacement of “range” with “scale”
- line 291: stylistic correction
- line 316: deletion of repeated definition of acronym
- line 323: stylistic correction
- line 333: deletion of repeated definition of acronym
- line 349: stylistic correction
- line 359: deletion of full name for acronym defined before
- line 434: stylistic correction
- line 436 (table 1) change of DFOS to DFOs; change of possibility of strain measurement at strain-hardening state with Polyimide to “-“ since it was not proven in the current work.
- lines 440, 442, 448.458 466-468,471,473: stylistic correction
- lines 481-501: section Conclusions re-written to underline the novelty of the current research thus use of DFOs for UHPFRC to detect and observe the cracks/discontinuities. Range of use of two kinds of fiber optic sensors emphasised. Possibility of measurement of opening-closing crack highlighted.
Reviewer 2 Report
This manuscript has a really strong body, with adequate experiment measurements and results, but a poorly written introduction and conclusion sections.
The manuscript is very incoherent, and is clearly taken from a book manuscript; seeing that the author forgot to remove the reference: "In this chapter", and "following chapters". The style of writing in the body is not that for an article, but for a book chapter, clearly again, cut and pasted from another book manuscript of this group.
Acronyms should be defined only once in the intro and body of the article. Remove the multiple definition of acronyms in lines 60 and 62 and 67 and throughout the rest of the body. Some acronyms like DFOs and UHPFRC are defined differently than their what their known for.
The manuscript needs a through formatting and punctuation revision. Intro is not aligned like the rest of the text. Line 100: remove the comma at the end.
The manuscript needs an extensive editing of English language and style. The introduction is very poorly written.
Line 113: Explain what G and E are.
The second and third conclusions are very weakly described. The optical fibers on their own won't allow for strain measurements. It's the whole distributed strain sensor system with the optical fibers and the OBR interrogator that allow for such measurements to be attained. Naming specific fibers is inaccurate, most other germanosilicate fibers with polyimide or even acrylic coating will do the job. These results are not new. The author is advised to re-write the conclusions section.
Referencing 42 source for this extremely ubiquitous topic, in a 20-page manuscript, is simply in-sufficient.
Author Response
The authors would like to express their gratitude to reviewers for their time spent on this task. Their comments and suggestions were very useful and undoubtedly will help in improvement of the manuscript.
The detailed responses to the comments are listed below. The word file was edited with change tracking to ease-up further stages of verification. The modifications are listed below.
This manuscript has a really strong body, with adequate experiment measurements and results, but a poorly written introduction and conclusion sections.
The introduction and conclusions sections have been modified from linguistic and stylistic points of view (see list of modifications for details)
The manuscript is very incoherent, and is clearly taken from a book manuscript; seeing that the author forgot to remove the reference: "In this chapter", and "following chapters". The style of writing in the body is not that for an article, but for a book chapter, clearly again, cut and pasted from another book manuscript of this group.
The current manuscript is not intended to be published, nor has been published as part of a book manuscript. The references: “In this chapter” and “following chapters” referred to relevant sections of the paper. To avoid this misunderstanding, the whole linking paragraph is removed.
Acronyms should be defined only once in the intro and body of the article. Remove the multiple definition of acronyms in lines 60 and 62 and 67 and throughout the rest of the body. Some acronyms like DFOs and UHPFRC are defined differently than their what their known for.
The repeated definitions of acronyms are deleted.
The use of UHPFRC acronym for Ultra High Performance Fibre Reinforced Cementitious composites (or Ultra High Performance Fibre Reinforced Concrete, depending on author) is well established in the literature (e.g.: Designing and Building with UHPFRC, François Toutlemonde Jacques Resplendino, 2011; "Structural UHPFRC": Welcome to the post-concrete era!; Eugen Brühwiler, keynote lecture at UHPC Symposium, Des Moines, Iowa, USA, July 18-20, 2016)
The use of DFOs (or DFOS, DOFs depending on author) is an established acronym for Distributed Fibre Optics sensors (or sensing), e.g.: The monitoring of an existing cast iron tunnel with distributed, Gue et al., 2015; Integrity Testing of Pile Cover Using Distributed Fibre Optic Sensing, Rui et al., 2017.
The manuscript needs a through formatting and punctuation revision. Intro is not aligned like the rest of the text. Line 100: remove the comma at the end.
The formatting of introduction is corrected, coma removed. The article is proof-read to verify the formatting; see list of modifications for details.
The manuscript needs an extensive editing of English language and style. The introduction is very poorly written.
The introduction and the rest of body of the manuscript is edited. See the list of modifications for details.
Line 113: Explain what G and E are.
Symbols explained.
The second and third conclusions are very weakly described. The optical fibers on their own won't allow for strain measurements. It's the whole distributed strain sensor system with the optical fibers and the OBR interrogator that allow for such measurements to be attained. Naming specific fibers is inaccurate, most other germanosilicate fibers with polyimide or even acrylic coating will do the job. These results are not new. The author is advised to re-write the conclusions section.
Conclusions section is re-written with relevant observations underlined.
Referencing 42 source for this extremely ubiquitous topic, in a 20-page manuscript, is simply in-sufficient.
Additional references are added for strain transfer in optical sensors and use of optical sensors in SHM. There is no literature on use of fibre optical sensors for crack measurements in UHPFRC - this fact was the main motivation towards the presented research.
List of modifications (line numeration of edited article used):
- Whole document: re-ordering of references according to modifications is text
- line 13: typo corrected
- lines24-30: stylistic corrections
- lines 34-68: stylistic corrections
- lines 49-51: sentence added to show novelty of research
- lines 61-63: added reference [12] with use of DFOs for SHM in UHPFRC structure
- line 70: deletion of abbreviation form section title
- lines 71-74: stylistic corrections
- line 94: stylistic corrections
- lines 103-110: addition of references [29-37] on strain transfer in optical fiber sensor
- lines 112,116: stylistic corrections
- line 121: definition of G and E added
- lines 125-127,132,133,135,137: stylistic corrections
- line 143: replacement of abbreviation with full name in the section title
- lines 210, 211,219, 220: stylistic corrections
- line 227: deletion of linking paragraph
- line 232: deletion of “in the following chapters” to avoid misunderstandings
- line 271 (caption fig. 9): replacement of “range” with “scale”
- line 291: stylistic correction
- line 316: deletion of repeated definition of acronym
- line 323: stylistic correction
- line 333: deletion of repeated definition of acronym
- line 349: stylistic correction
- line 359: deletion of full name for acronym defined before
- line 434: stylistic correction
- line 436 (table 1) change of DFOS to DFOs; change of possibility of strain measurement at strain-hardening state with Polyimide to “-“ since it was not proven in the current work.
- lines 440, 442, 448.458 466-468,471,473: stylistic correction
- lines 481-501: section Conclusions re-written to underline the novelty of the current research thus use of DFOs for UHPFRC to detect and observe the cracks/discontinuities. Range of use of two kinds of fiber optic sensors emphasised. Possibility of measurement of opening-closing crack highlighted.
Round 2
Reviewer 1 Report
The changes performed by the authors was significant improved the quality of the paper.
However, for some reason the figures were not included in the manuscript. Since some of the comments are related with the Figures, I can not suggest the acceptance of the paper, yet.
Author Response
Dear reviewers,
the figures were not attached in the last version of the document due to misunderstanding between authors and editorial office, sorry for that.
The only modification in the current Revision 2 comparing with Revision 1 is addition of the figures to the body of document and removal of typos in lines 1 and 291.
The figures are unedited comparing to the original version of the manuscript.
Again, thank you for your comments and time spent on the review.
Reviewer 2 Report
The revised manuscript is highly improved. The figures are missing in the manuscript pdf though, I had to compare to the old manuscript.
Author Response

(The authors gave the same response as above.)
